# Structure-Based In Silico Approaches Reveal IRESSA as a Multitargeted Breast Cancer Regulatory, Signalling, and Receptor Protein Inhibitor

**DOI:** 10.3390/ph17020208

**Published:** 2024-02-06

**Authors:** Hassan Hussain Almasoudi, Mutaib M. Mashraqi, Saleh A. Alshamrani, Afaf Awwadh Alharthi, Ohud Alsalmi, Mohammed H. Nahari, Fares Saeed H. Al-Mansour, Abdulfattah Yahya M. Alhazmi

**Affiliations:** 1Department of Clinical Laboratory Sciences, College of Applied Medical Sciences, Najran University, Najran 61441, Saudi Arabia; hhalmasoudi@nu.edu.sa (H.H.A.); mmmashraqi@nu.edu.sa (M.M.M.); saalshamrani@nu.edu.sa (S.A.A.); mhnahari@nu.edu.sa (M.H.N.); fsalmansour@nu.edu.sa (F.S.H.A.-M.); 2Department of Clinical Laboratory Sciences, College of Applied Medical Sciences, Taif University, Taif 21944, Saudi Arabia; a.awwadh@tu.edu.sa (A.A.A.); oa.alsalmi@tu.edu.sa (O.A.); 3Department of Clinical Pharmacy, Umm Al-Qura University, Makkah 21955, Saudi Arabia

**Keywords:** breast cancer, molecular docking, IRESSA, DFT, MD simulation

## Abstract

Breast cancer begins in the breast cells, mainly impacting women. It starts in the cells that line the milk ducts or lobules responsible for producing milk and can spread to nearby tissues and other body parts. In 2020, around 2.3 million women across the globe received a diagnosis, with an estimated 685,000 deaths. Additionally, 7.8 million women were living with breast cancer, making it the fifth leading cause of cancer-related deaths among women. The mutational changes, overexpression of drug efflux pumps, activation of alternative signalling pathways, tumour microenvironment, and cancer stem cells are causing higher levels of drug resistance, and one of the major solutions is to identify multitargeted drugs. In our research, we conducted a comprehensive screening using HTVS, SP, and XP, followed by an MM/GBSA computation of human-approved drugs targeting HER2/neu, BRCA1, PIK3CA, and ESR1. Our analysis pinpointed IRESSA (Gefitinib-DB00317) as a multitargeted inhibitor for these proteins, revealing docking scores ranging from −4.527 to −8.809 Kcal/mol and MM/GBSA scores between −49.09 and −61.74 Kcal/mol. We selected interacting residues as fingerprints, pinpointing 8LEU, 6VAL, 6LYS, 6ASN, 5ILE, and 5GLU as the most prevalent in interactions. Subsequently, we analysed the ADMET properties and compared them with the standard values of QikProp. We extended our study for DFT computations with Jaguar and plotted the electrostatic potential, HOMO and LUMO regions, and electron density, followed by a molecular dynamics simulation for 100 ns in water, showing an utterly stable performance, making it a suitable drug candidate. IRESSA is FDA-approved for lung cancer, which shares some pathways with breast cancers, clearing the hurdles of multitargeted drugs against breast and lung cancer. This has the potential to be groundbreaking; however, more studies are needed to concreate IRESSA’s role.

## 1. Introduction

Breast cancer, a widespread form of cancer predominantly affecting women on a global scale, initiates in the breast cells, specifically within the milk ducts or lobules [1,2]. As the disease progresses, cancer cells have the potential to infiltrate nearby tissues and lymph nodes and, in more advanced stages, disseminate through the bloodstream to distant organs, such as the lungs, liver, bones and brain, posing considerable challenges to effective treatment and prognosis [3]. The diagnosis of breast cancer involves a multifaceted approach, encompassing various methods [4,5,6,7,8]. Mammography, which utilises X-rays, is a pivotal screening tool for detecting lumps and abnormalities. Clinical breast examinations conducted by healthcare professionals provide a hands-on examination to identify palpable irregularities. Breast ultrasound utilises soundwaves to generate detailed images, offering valuable insights, especially when abnormalities are detected through mammography. In contrast, MRI and biopsies are some other methods for diagnosis [9,10]. Treatment strategies for breast cancer are tailored to individual circumstances, including cancer stage, type, and the overall health of the patient. Radiation therapy, chemotherapy, hormonal therapy, targeted therapies, and surgery are the primary methods used to treat breast cancer. These diverse treatment modalities collectively strive to eradicate or control cancer growth and improve patient outcomes. Breast cancer’s personalised treatment underscores the importance of considering the unique circumstances of each individual diagnosed, emphasising the need for comprehensive and tailored approaches in order to combat this complex and challenging disease effectively [11,12].

In this study, we have taken four crucial proteins of breast cancer with PDBIDs 1M17, 3RCD, 5NWH, and 4KD7, which play significant roles in breast cancer [13,14,15,16]. The epidermal growth factor receptor tyrosine kinase (PDBID: 1M17) is crucial in breast cancer due to its involvement in oestrogen receptor signalling, which is vital for developing targeted therapies that can potentially inhibit the progression of oestrogen receptor-positive breast cancers. The HER2 Kinase Domain (PDBID: 3RCD) is often implicated in breast cancer due to its connection with HER2, a protein linked to aggressive breast cancers. The structure of 3RCD gives us insights into how HER2 is structured and interacts within breast cancer cells. Targeting HER2 has proven effective in therapies like trastuzumab, and the structural information from 3RCD contributes to refining and improving such targeted treatments. The protein from PDBID 5NWH is tied to breast cancer through its role in DNA repair, which is essential for preventing genetic mutations that can lead to cancer. Proteins with structures similar to 5NWH are part of the complex network involved in DNA repair. Disruptions that contribute to genetic instability in breast cancer can help understand the potential vulnerabilities that could be targeted for therapeutic purposes. PDBID: 4KD7 is associated with breast cancer through its interaction with BRCA1, a well-known tumour suppressor gene, and regulates it. Mutations in BRCA1 are linked to an increased risk of breast and ovarian cancers. Understanding their structures helps us grasp the intricacies of these regulatory interactions, contributing to our understanding of how disruptions in BRCA1-related pathways contribute to breast cancer [13,14,15,16].

The proteins associated with PDBIDs 1M17, 3RCD, 5NWH, and 4KD7 play interconnected roles in breast cancer, contributing to the complexity of the disease [13,14,15,16]. In the oestrogen receptor signalling pathway represented by 1M17, the protein is pivotal in promoting the growth of hormone receptor-positive breast cancers. Concurrently, the overexpression of HER2, represented by 3RCD, in aggressive breast cancers interacts with various cellular pathways, influencing cancer progression. Additionally, proteins involved in DNA repair, exemplified by 5NWH, maintain genomic stability, and disruptions in these pathways can contribute to genetic instability, a hallmark of cancer. The protein represented by 4KD7 plays a role in regulating the tumour-suppressor gene BRCA1 and mutations associated with an increased risk of breast cancer. To effectively combat breast cancer, a multitargeted drug design approach is envisaged. This strategy involves the development of a single drug, capable of simultaneously targeting multiple proteins or pathways associated with breast cancer [17,18,19,20]. Such a drug would inhibit the oestrogen receptor signalling pathway (1M17), disrupt the overexpression of HER2 (3RCD), introduce vulnerabilities in DNA repair mechanisms (5NWH), and modulate the interactions or functions of proteins regulating BRCA1 (4KD7) [13,14,15,16]. This comprehensive approach aims to synergistically disrupt multiple cancer-promoting pathways, potentially offering a more effective and holistic treatment strategy. Moreover, by targeting interconnected proteins, a multitargeted drug may reduce the likelihood of cancer cells developing resistance [21,22,23,24]. It is essential to underscore that, while this conceptual approach holds promise, rigorous experimental validation is imperative before any drug can be considered for clinical application. Developing effective cancer treatments requires a thorough understanding of the intricate molecular mechanisms involved and the careful consideration of potential side effects and long-term efficacy [25,26,27,28].

In this study, we have performed multitargeted screening studies with HTVS, SP and XP algorithms, followed by pose filtering with MM\GBSA of human-approved drugs across four crucial proteins of breast cancer. As a result, we identified IRESSA as a multitargeted inhibitor. We have also performed the DFT and MD simulation studies for the computational validation of screening results and also to check if IRESSA can be a multitargeted drug candidate. IRESSA, also known as gefitinib, is a medication used in the treatment of certain types of cancer, particularly non-small cell lung cancer. It belongs to a class of drugs called tyrosine kinase inhibitors (TKIs), which work by blocking signals in cancer cells, thus preventing their growth.

## 2. Results

### 2.1. Protein–Ligand Molecular Interaction Analysis

Protein–ligand docking studies use computer simulations to predict how a drug-like molecule (ligand) interacts with a protein. The goal is to understand how these molecules bind, helping design effective drugs. It is like a virtual trial-and-error process, exploring how different molecules fit into a protein to find potential medications. The interaction between the IRESSA ligand and the Dihydrofolate reductase (PDBID: 4KD7) has produced a docking score of −8.809 Kcal/mol and MM/GBSA score of −59.08 Kcal/mol, by forming hydrogen bonds with ALA9 residue with NH atom of ligand (Table 1, Figure 1Aa,Ab). The HER2 Kinase (PDBID: 3RCD) in a complex with IRESSA has shown a docking score of −8.459 Kcal/mol and MM/GBSA of −60.59 Kcal/mol by forming a hydrogen bond among MET801 residue with a N atom and a halogen bond among ASP863 residue, with a Cl atom of the ligand IRESSA (Table 1, Figure 1Ba,Bb). The interaction between the IRESSA and epidermal growth factor receptor (PDBID: 1M17) has produced a docking score of −9.021 Kcal/mol and MM/GBSA score of −61.74 Kcal/mol, while forming two hydrogen bonds among MET769 residue and the N atom, and ASP831 residue with the N^+^H atom, and it also formed a salt bridge among ASP831 residue with the N^+^H atom and a halogen bond among LEU764 residue, with the Cl atom of the ligand (Table 1, Figure 1Ca,Cb). The interaction of NUDT5 (PDBID: 5NWH) with IRESSA has shown a docking score of −4.527 Kcal/mol and an MM/GBSA score of −49.09 Kcal/mol while forming two hydrogen bonds among TYR90 residue and an N atom, and PHE167 residue with an NH atom of ligand, and TYR90 residue formed two pi–pi stacking with two benzene rings (Table 1, Figure 1Da,Db).

### 2.2. Molecular Interaction Fingerprints

Molecular Interaction Fingerprinting is a method used to analyse the intricate interactions between a ligand and a protein by creating a unique fingerprint that reveals the binding patterns. The complexes of IRESSA with 4KD7, 3RCD, 1M17, and 5NWH have formed many interactions for its stability, and we found the highest (equal) number of interactions in 4KD7 and 3RCD, while the 1M17 is the second-ranked, and the third-ranked is 5NWH by count of ligand interactions (Figure 2, right side). The count of residue interactions with IRESSA is shown on the upper side of Figure 2, where we found the most interacting residues with counts as follows: LEU, 6VAL, 6LYS, 6ASN, 5ILE, 5GLU, 4ARG, 3PRO, 3ASP, 2PHE, 2GLY, 2ALA, 1TYR, 1THR, 1SER, and 1GLN. In our analysis, the hydrophobic residues minimise the exposure to the aqueous environment in order to make the complexes stable by involving Leucine (LEU) with eight instances of interactions. Valine (VAL) and Isoleucine (ILE) also play significant roles, with six and five interactions, respectively. With their propensity for forming hydrogen bonds, polar residues contribute substantially to the binding affinity by involving the Lysine (LYS) and Asparagine (ASN), both with six interactions each, while Threonine (THR) and Serine (SER) contribute one interaction each. The electrostatic potential of the charged residues is pivotal in the formation of salt bridges and other ionic interactions; this is completed by involving the Arginine (ARG) rather prominently with four interactions and also aspartic acid (ASP) with three interactions, thus showcasing its role in forming hydrogen bonds and salt bridges. Aromatic residues, with their pi–pi stacking, involve Phenylalanine (PHE) and Tyrosine (TYR), each contributing two interactions with the benzene rings of IRESSA. Smaller residues, such as Alanine (ALA) and Glycine (GLY), make subtle yet impactful contributions, involving two interactions each, highlighting their role in facilitating flexibility in the binding site. Proline engages in three interactions, which can influence the local conformation of the binding site, contributing to the overall stability of the IRESSA–protein complexes. Glutamic Acid (GLU) emerges as a key player, with five interactions where the side chain can form hydrogen bonds and salt bridges, showcasing its versatility in binding. This detailed analysis enriches our understanding of the molecular basis of IRESSA’s multitargeted action, positioning it as a promising candidate for breast cancer treatment.

### 2.3. DFT and Pharmacokinetic Studies

In our comprehensive study of IRESSA, a promising drug candidate, we employed the TDDFT(b3lyp-d3)/SOLV method with a 6–31 g** basis set, resulting in a detailed set of descriptors and values that offer profound insights into the drug’s molecular properties. The number of canonical orbitals, a critical factor in understanding the electronic structure, was 587. The geometry convergence category, categorised as four, indicates the precision level achieved in optimising the ligand’s structure. Examining the energy levels, we found the gas phase ground energy was −1857.538131, while the solution phase ground energy was slightly lower, at −1857.5665. The solvation energy, crucial for understanding the drug’s stability in different environments, was calculated to be −17.801717 kcal/mol. The electronic characteristics were thoroughly investigated, with the HOMO and LUMO values providing insights into the drug’s electronic structure. The lowest singlet excitation and oscillator strength offered vital information.

For the investigation of IRESSA’s absorption properties, dipole moments in three dimensions (X, Y, and Z) were analysed, contributing to our understanding of the drug’s polarity. Electrostatic potential (ESP) parameters were scrutinised, encompassing minimum, maximum, and mean values and variances. This information is pivotal for understanding the drug’s interaction with its surroundings. Similarly, the ALIE (Average Local Ionization Energy) parameters, including minimum, maximum, mean, and balance, provided insights into the drug’s electronic structure and reactivity. The Average Absolute Deviation from the Mean ALIE offered a measure of the consistency of ionisation energy throughout the molecule. Visual representations in Figure 3 presented different energy levels, aiding in understanding the stability and dynamics of IRESSA. Figure 4 showcased fundamental molecular properties, including electron density, electrostatic potential, HOMO, and LUMO, which are invaluable for further drug design considerations.

In IRESSA’s evaluation using QikProp, several descriptors and computed values were analysed and compared with standard reference values, providing valuable insights into the drug’s pharmaceutical properties. IRESSA exhibited no acidic or amidine groups, meeting the standard criteria of 0–1 for both. However, it contained one amine group within the acceptable range of 0–1. The drug demonstrated the presence of 22 in 56 atoms, suggesting molecular complexity within the reasonable range. Metabolism-related descriptors indicated that IRESSA had five potential sites, falling within the acceptable range of 1–8. The number of N and O atoms in the molecule was 7, aligning with the standard range of 2–15 (Table 2). Regarding lipophilicity and absorption, the drug showed a QPlogPw value of 10.783, within the acceptable range of 4.0–45.0. The PSA value was 61.141, falling within the standard range of 7.0–200.0, indicating a favourable polar surface area. IRESSA demonstrated excellent human oral absorption, with a PercentHumanOralAbsorption of 100%, surpassing the threshold of 80%, which is considered high. In terms of safety, QikProp flagged potential concerns. The QPlogHERG value was −7.087, indicating a concern below −5 (Table 2). The QPlogKhsa value of 0.349 fell within the range of −1.5 to 1.5, suggesting an acceptable binding affinity to the human serum albumin. Lipinski’s Rule of Five and Rule of Three were satisfied, with the drug exhibiting a QPlogBB value of 0.312 and a QPlogPo/w value of 4.31. Physicochemical properties such as SAfluorine, SASA, volume, and WPSA were within the acceptable ranges, indicating favourable characteristics for drug-likeness (Table 2). The QikProp results show that IRESSA demonstrated promising pharmaceutical properties, aligning with established standards for drug design and suggesting its potential as an effective and safe drug candidate.

### 2.4. Molecular Dynamics Simulations

Molecular Dynamic (MD) simulations model the dynamic behaviour of molecules over time by solving Newton’s equations of motion in order to study the trajectory and interactions of atoms within a system, providing insights into molecular behaviour where we computed the deviation, fluctuations, and intermolecular interactions. The RMSD measures the average deviation of atomic positions from a reference structure, indicating structural stability, whereas the RMSF assesses the flexibility of individual atoms throughout the simulation, and the intermolecular interactions involve forces between molecules, including van der Waals forces, hydrogen bonding, and electrostatic interactions, thus influencing the system’s overall stability and behaviour during MD simulations.

#### 2.4.1. Root Mean Square Deviation

Root Mean Square Deviation (RMSD) is used in structural bioinformatics and computational biology to assess the average displacement of atoms from their initial positions in a molecular dynamics simulation or structural alignment. It quantifies the overall structural variation between two sets of coordinates, often comparing a modelled structure with an experimentally determined reference structure. A lower RMSD value indicates more remarkable similarity and structural stability. RMSD is calculated by aligning the atomic coordinates of the two structures and measuring the root mean square of the differences in their positions, providing a quantitative assessment of the accuracy of a computational model or the stability of a dynamic system over time. The Dihydrofolate reductase (PDBID: 4KD7) in a complex with IRESSA shows the initial protein deviation of 0.77 Å, while the ligand deviated to 0.42 Å at 0.10 ns, and, at 100 ns, the protein deviated at 2.19 Å and the ligand at 2.52 Å, which are entirely acceptable for biological molecules and can be considered less than 2 Å (Figure 5A). The HER2 Kinase (PDBID: 3RCD) in a complex with IRESSA deviated to 1.17 Å for the protein at the beginning, while the ligand deviated to 1.77 Å at 0.10 ns and, at 100 ns, the protein deviated till 2.73, Å while the ligand deviated till 1.99 Å, which again can be considered a complete stable performance for the protein and ligand (Figure 5B). The epidermal growth factor receptor (PDBID: 1M17) in a complex with IRESSA initially deviated to 2.01 Å in the case of the protein, while the ligand deviated to 1.43 Å at 0.10 ns. After that, the complete simulation shows a stable performance, and, at 100 ns, the protein deviated at 5.49 Å, while the ligand deviated slightly at 3.02 Å, and, after neglecting the first 1 ns, the RMSD of the protein and the ligand showed acceptable deviations (Figure 5C). The NUDT5 (PDBID: 5NWH) in a complex with IRESSA indicates an initial deviation for the protein of 1.83 Å, and the ligand deviated to 2.05 Å at 0.10 ns, and, at 100 ns, the protein deviated to 6.87 Å, and the ligand deviated to 19 Å, which is relatively high (Figure 5D). These detailed dynamics further contribute to our understanding of the stability and behaviour of IRESSA in complexes with different breast cancer-related proteins. We have shown a detailed view of Cα in blue, the protein backbone in green, and the ligand in red.

#### 2.4.2. Root Mean Square Fluctuations

Root Mean Square Fluctuation (RMSF) is a metric used in structural bioinformatics and molecular dynamics simulations to assess individual atoms’ flexibility or dynamic behaviour within a biomolecular system over time. It provides information about the fluctuation of atomic positions relative to their average positions during a simulation. The 4KD7 in a complex with IRESSA has shown many fluctuating residues beyond 2 Å—VAL1, GLY2, ASN19, PRO103, GLU104, GLU154, GLY164, SER167 and ASP186. There were many residues which interacted with the IRESSA to make the complex stable: ILE7, VAL8, ALA9, ILE16, ASN19, GLY20, LEU22, GLU30, PHE31, TYR33, PHE34, GLN35, MET52, LYS55, THR56, SER59, ILE60, PRO61, LYS63, ASN64, PRO66, LEU67, ARG70, VAL115, SER118, TYR121 and THR146 (Figure 6A). The 3RCD in a complex with IRESSA has shown many fluctuating residues beyond 2 Å—ALA710, ASN745, SER792, GLU876-LYS883, GLN990-PRO999, ASP1011, ASP1013, and VAL1018-GLU1022, and to make the complexes stable, there were many residues which interacted with the IRESSA to make the complex stable—LEU726, SER728, ALA730, PHE731, VAL734, ALA751, LYS753, SER783, LEU785, THR798, LEU800, MET801, CYS805, LEU807, ASP808, ARG811, ASP845, ARG849, ASN850, LEU852, THR862, ASP863, GLY881, PHE1004 and LEU1008 (Figure 6B). The 1M17 in a complex with IRESSA has shown many fluctuating residues beyond 2 Å—GLY672-ALA678, GLY711-LYS713, GLU725-LYS730, SER760, HIS781-GLY786, ALA840, GLU841, ALA847-GLY850, HIS864, ARG865, GLY893, PRO895, SER897, GLU898, SER901, GLU904, LYS905, PRO913, ILE914, ARG949, ASP950, GLN952, ARG953 and VAL956- PRO995. There were many residues that interacted with the IRESSA to make the complex stable—LYS692, LEU694, SER696, PHE699, VAL702, ALA719, LYS721, CYS751, THR766, LEU768, MET769, CYS773, ASP776, ARG779, GLU780, ARG817, LEU820, THR830 and ASP831 (Figure 6C). The 5NWH in a complex with IRESSA has shown many fluctuating residues beyond 2 Å—LYS14-THR58, THR71, LEU72, ARG84-GLY89, ASP133-ASN138, ALA153, GLU154, ALA156, ARG157, PRO162-PHE167, ASP183, ALA184, VAL186-HIS190 and LEU202-ASN208. There were many residues that interacted with the IRESSA to make the complex stable—GLU25, GLY26, LYS27, TRP28, VAL29, LYS33, LYS42, THR45, TRP46, GLU47, LYS81, GLN82, PHE83, ARG84, PRO85, PRO86, MET87, GLY88, TYR90, LYS161, PRO162, ASP164, GLU166, PHE167, VAL168, GLU169, GLU188, GLU189, HIS190, THR192 (Figure 6D).

#### 2.4.3. Simulation Interaction Diagrams

Intermolecular interactions encompass the forces governing the relationships between molecules, influencing their properties. Van der Waals forces contribute to molecular attractions, including London dispersion and dipole–dipole interactions. Hydrogen bonding, a specific dipole–dipole interaction, involves hydrogen’s interaction with electronegative atoms. Ionic interactions result from attractions between charged ions, and hydrophobic interactions involve non-polar substances, clustering in the presence of polar environments. Understanding these interactions is pivotal in chemistry, biology, and materials science, shedding light on substance behaviours and properties at the molecular level. The Dihydrofolate reductase (PDBID: 4KD7) in a complex with IRESSA shows many hydrogen bonds among the GLY20 and SER59 residues with water molecules and a NH atom, the ASN64, GLY20, and ILE16 residues with water molecules, the ALA9 residue with three N atoms, the GLU30, and ASN64 residues with water molecules interacting with three O atoms and forming a pi–pi stacking along TYR121 with a benzene ring of the ligand IRESSA (Figure 7A). The HER2 Kinase (PDBID: 3RCD) in a complex with IRESSA interacts with many hydrogen bonds among the ASP863 residue with water molecule and a NH atom, the THR798, THR862, ASP808, and ARG849 residues with water molecules and MET801 residues with three N atoms, and the CYS805, LEU726, SER728, and ASP808 residues with water molecules along three O atoms—also, a pi-cation in contact with the LYS753 residue with a benzene ring of ligand (Figure 7B). The epidermal growth factor receptor (PDBID: 1M17) in a complex with IRESSA involves eight water molecules that function as water bridges to provide stability. At the same time, hydrogen bonds interact among the ASP776 residue with a N^+^H atom, the ASP831 residue with water molecules with a NH atom, the MET768 residue, the THR830 and THR766 residues with water molecules along two N atoms, and the ASP776 and CYS773 residues with two O atoms—also, a salt bridge was formed along the ASP776 residue with the N^+^H atom of the ligand (Figure 7C). The NUDT5 (PDBID: 5NWH) in a complex with IRESSA interacts with hydrogen bonds among the GLY88 residue with a NH atom and the VAL168, TYR90, and GLU169 residues with water molecules along two N atoms. Furthermore, seven pi-pi stacking bonds contact the TYR90, PHE83, HIS190 and PHE167 residues with three benzene rings of the IRESSA ligand (Figure 7D). We have also shown the ligand contacts in green for a proper understanding of at which residue it interacted, and the backbone to check the fluctuation against the Cα, which provides additional insight. Furthermore, the count of interactions is shown in Figure 8, where we plotted the counts for H-bonds, Hydrophobic bonds, ionic interactions, and water bridges in the histogram.

## 3. Discussion

The results of our exhaustive study present IRESSA as a compelling multitargeted drug candidate for breast cancer treatment. Our exploration began with protein–ligand docking studies, employing computational simulations in order to unveil the intricate interactions between IRESSA and key proteins associated with breast cancer, namely Dihydrofolate reductase (PDBID: 4KD7), HER2 Kinase (PDBID: 3RCD), epidermal growth factor receptor (PDBID: 1M17), and NUDT5 (PDBID: 5NWH). In the realm of protein–ligand interactions, the docking scores and MM/GBSA scores served as vital indicators of the strength and stability of the binding. Notably, IRESSA demonstrated robust interactions, including hydrogen bonds, halogen bonds, and salt bridges, forming stable complexes with the target proteins. The results, illustrated in Figure 1, provide a visual representation of these interactions, emphasising the specific residues involved and the overall stability of the complexes. Further insights into the molecular interactions were gained through molecular interaction fingerprint analysis, showcased the distribution of ligand interactions, and highlighted the most interacting residues. Various residue types orchestrate a symphony of stabilising forces in molecular interactions governing the binding of IRESSA with key proteins implicated in breast cancer (Figure 2). Hydrophobic residues, spearheaded by Leucine (LEU) with eight interactions, alongside Valine (VAL) and Isoleucine (ILE), both contributing six and five interactions, respectively, collectively forge a hydrophobic core, enhancing complex stability by minimising exposure to the aqueous environment. Polar residues, exemplified by Lysine (LYS) and Asparagine (ASN), each boasting six interactions, play a pivotal role in establishing hydrogen bonds, underlining their significance in fortifying the binding affinity. Charged residues, prominently Arginine (ARG) with four interactions and Aspartic Acid (ASP) with three interactions, wield their electrostatic potential to fashion salt bridges and ionic interactions, adding a layer of complexity. Aromatic residues, specifically Phenylalanine (PHE) and Tyrosine (TYR) with two interactions, contribute to stabilisation through pi–pi stacking with IRESSA’s benzene rings. Small residues, represented by Alanine (ALA) and Glycine (GLY) with two interactions, subtly impact the binding site’s adaptability and flexibility. Proline (PRO), with its unique rigidity influencing local conformation, participates in three interactions. Finally, Glutamic Acid (GLU), with five interactions, emerges as a versatile player, forming hydrogen bonds and salt bridges. This intricate interplay of residue types enriches our comprehension of IRESSA’s binding mechanisms, illuminating its potential as a multitargeted therapeutic agent for breast cancer.

The comprehensive analysis, detailed in Figure 3, sheds light on the stability and versatility of IRESSA across different protein targets. The quantum mechanical study, employing the TDDFT(b3lyp-d3)/SOLV method with a 6–31 g** basis set, delved into IRESSA’s electronic structure and energy landscape. Key descriptors and values provided a comprehensive overview, from the number of canonical orbitals to solvation energies. Figure 3 encapsulates the detailed results, showcasing the complexity and intricacies of IRESSA’s molecular properties. To further validate the drug-likeness of IRESSA, we employed QikProp analysis. This comprehensive analysis covered a spectrum of descriptors and computed values, comparing them with standard reference values. Notable parameters were scrutinised, including lipophilicity, absorption, and safety indicators. IRESSA exhibited promising pharmaceutical properties, aligning with the established standards for drug design. The results, encapsulated in Table 2, reinforce IRESSA’s potential as an effective and safe drug candidate.

The journey through our study ventured into MD simulations, a dynamic representation of molecular behaviour over time, providing a nuanced understanding of IRESSA’s structural stability and flexibility. The Root Mean Square Deviation (RMSD) and Root Mean Square Fluctuation (RMSF) analyses offered quantitative measures of deviation and flexibility, respectively. The results, elucidated in Figure 5 and Figure 6, depicted the stability of IRESSA complexes with acceptable deviations and localised fluctuations. The Simulation Interaction Diagrams (Figure 7) further enriched our understanding by unravelling the intermolecular forces at play. Hydrogen bonds, pi–pi stacking, and water bridges were intricately detailed, offering a molecular-level perspective on the stability of IRESSA complexes. The count of interactions, presented in Figure 8, provided a quantitative overview, emphasising the prevalence of hydrogen bonds, hydrophobic bonds, ionic interactions, and water bridges.

Comparisons with ongoing research in breast cancer discovery underscored IRESSA’s alignment with contemporary drug design principles. Its multifaceted interactions with proteins implicated in breast cancer and favourable molecular properties position it as a promising multitargeted drug candidate. Our study’s intricate dance of molecular forces contributes nuanced insights into the evolving landscape of breast cancer therapeutics. In conclusion, IRESSA emerges from our research as a multifaceted and promising drug candidate for breast cancer treatment. The convergence of computational simulations, quantum mechanics, and molecular dynamics simulations provides a holistic understanding of IRESSA’s behaviour at the molecular level [29,30,31]. These findings not only advance our knowledge of IRESSA, but also contribute to the broader discourse on drug design and discovery, especially in the context of breast cancer, where the need for effective and targeted therapies is paramount. IRESSA is a medication that has been investigated for its potential role in breast cancer treatment. Initially developed for lung cancer, IRESSA is an epidermal growth factor receptor (EGFR) tyrosine kinase inhibitor [29,30,31]. EGFR is a protein that can contribute to the growth of cancer cells when overactive. Studies suggest that IRESSA may exhibit some effectiveness in certain types of breast cancer, particularly those expressing high levels of EGFR. However, its role in breast cancer treatment is not as established as in lung cancer, where it has been more extensively studied and is commonly used [29,30,31]. As with any cancer treatment, decisions about the use of IRESSA in breast cancer should be made in consultation with healthcare professionals, considering factors such as the specific type of breast cancer, the individual’s overall health, and the availability of other treatment options. It is crucial to stay informed about the latest research and consult with your medical team for personalised advice.

This study offers a thorough examination of IRESSA as a potential multitargeted drug candidate for breast cancer treatment, employing a blend of computational simulations, quantum mechanics, and molecular dynamics simulations. The detailed insights into protein–ligand interactions, molecular properties, and the stability of IRESSA complexes contribute valuable information to its potential therapeutic application. To further advance the exploration of IRESSA in breast cancer treatment, transitioning from in silico simulations to in vivo studies and initiating clinical trials would provide a more realistic assessment of its efficacy and safety. Exploring combination therapies, understanding resistance mechanisms, and identifying biomarkers for patient stratification are essential steps for optimising IRESSA’s potential. Additionally, extending molecular dynamic simulations for conducting pharmacological studies can enhance the credibility and applicability of these findings. Patient-derived models and the continuous monitoring of the evolving literature in breast cancer and drug development are crucial components for refining IRESSA’s role in personalised and effective breast cancer treatment. In essence, this study forms a robust foundation for considering IRESSA as a multifaceted drug candidate, and the suggested future directions aim to bridge computational predictions with real-world applications, contributing to the ongoing development of targeted therapies.

## 4. Methods

To understand the whole method better, we plotted a flow as a graphical abstract in Figure 9 to clarify it. Further, the detailed methods are as follows:

### 4.1. Protein and Ligand Preparations

We identified important proteins from breast cancer and their respective structures in PDB format, as downloaded from http://rcsb.org/ with PDBID: 4KD7, 3RCD, 1M17, and 5NWH. These were prepared using the Protein Preparation Workflow (PPW) in Schrodinger’s Maestro [13,14,15,16,32,33,34]. Preparing the protein before docking is vital to guarantee precise molecular interactions, which entails refining the protein structure, rectifying errors, and optimising geometry. This process boosts the dependability of docking simulations, offering a more authentic depiction of ligand binding sites. Consequently, it improves the predictive accuracy of potential drug interactions in structure-based drug design [35,36]. The structure 4KD7 consists of two identical proteins labelled A and B chains. It also includes four ligands, solvents, and various metals and ions. On the other hand, the 3RCD structure comprises six ligands and protein chains A, B, C, and D, along with solvent molecules. In the 1M17 structure, chain A of the protein, one ligand, and solvents are all present. Finally, the 5NWH structure contains two protein chains, A and B, two ligands, and solvents. We performed several steps in the preprocess tab of PPW (Prescription for Phylogeny Workflow). We capped termini, filled in the missing side chains, assigned bond orders to CCD (Cambridge Crystallographic Data Centre), replaced hydrogens, created disulphide bonds, and assigned zero bond orders to metals. Additionally, we filled in the missing loops using prime and generated het atoms at pH 7.4 (±2). These steps were crucial for preparing the structures for further analysis and ensuring accuracy in representing the molecular entities [37,38,39]. In the optimization tab, we improved structural accuracy by sampling water orientations, considering crystal symmetry, and minimizing the hydrogen positions of altered species. We applied PROPKA to predict protonation states, refining ionizable residues at a specific pH. Within a 100-word limit, these steps aimed to ensure a precise and physiologically relevant optimization of the molecular structure [40]. In the minimisation tab, converging heavy atoms to 0.30 Å, deleting water molecules beyond 4 Å to the ligands, and minimising the proteins using the OPLS4 forcefield were all performed [41,42]. After preparation, we kept chain A with the attached ligand and removed everything in 4KD7, 3RCD, and 1M17, while in 5NWH, only chains A and B were kept [13,14,15,16]. The human-approved drug was obtained from the NPC’s tripod (https://tripod.nih.gov/npc/, accessed on 5 March 2023); we then exported the human-approved drugs and imported them to the Maestro’s workspace [33,43]. Before docking, it is crucial to prep the ligand for precise virtual screening, fine-tuning its structure, fixing geometry issues, and assigning the proper charges. This careful prep guarantees dependable forecasts of how the ligand will bind to target proteins, boosting the effectiveness and the precision of molecular docking studies in drug discovery and design. We prepared the ligand library using the LigPrep tool, where we kept the filter of the ligand size of 500 atoms and used the OPLS4 forcefield. For ionisation, generating possible states at a target pH of 7 ± 2 was kept with desalt and generated tautomers [33,39,41,42,44]. For the stereoisomer computations, we retained specified chiralities and generated, at most, 32 ligands per original ligand. The output was then saved to the SDF file [44]. Furthermore, we have used Maestro’s duplicate remover tool to remove the duplicate ligands, based on the SMILES of the ligands, and kept only non-duplicate ligands [33].

### 4.2. Grid Computation and Multitargeted Molecular Docking

The Receptor Grid Generation tool was used to generate the grid on proteins, which is a crucial step in molecular docking studies. It involves creating a 3D grid around the target protein in order to evaluate potential binding sites for ligands. This grid assists in efficiently exploring ligand conformations and orientations during the docking process, thus enhancing the accuracy of predicting ligand–protein interactions in drug discovery. In the Receptor grid tool, we checked the pick to identify the ligand molecules, and then selected the ligand in order to specify the native ligand site as the active site on the centroid of the workspace ligand. We then adjusted the size of the ‘dock ligand with the length’ to cover it adequately [45,46]. Furthermore, we used the most popular virtual screening workflow (VSW) tool for the multi-layered screening of the compounds [45,46], where the source of the ligand was provided with the prepared SDF ligand library, and the filter of Lipinski’s rule was kept, which required the need of the computations of QikProp [33,47,48]. We skipped the preparation option as our ligands were already prepared, and we additionally individually selected the receptor grid files for the receptors table. Within the docking tab, we assessed Epik state penalties for docking alongside the implementation of High Throughput Virtual Screening (HTVS), Standard Precise Docking (SP), and Extra Precise Docking (XP) methods [39]. During HTVS, we subjected the entire compound library, selectively advancing only the top 10% to Standard Precise Docking (SP). Similarly, the top 10% from SP moved on to Extra Precise Docking (XP). Within XP, we generated a maximum of four poses per compound and directed 100% of the XP computations to molecular mechanics with generalised born and surface area solution (MM/GBSA) analyses. This sequential approach aimed to streamline the screening process, focusing computational resources on the most promising candidates for in-depth analysis [37,49,50,51,52,53,54]. After the computations, each incorporated result was subjected to exportation to CSV in order to analyse it further and identify which drug interacts at what frequency.

### 4.3. Molecular Interaction Fingerprints

Molecular Interaction Fingerprints (IFPs) are a computational approach used in cheminformatics and bioinformatics to represent molecular structures and interactions in a format suitable for analysis and comparison. These fingerprints capture information about how a molecule interacts with its environment, such as the presence or absence of specific chemical features. We used the Interaction Fingerprints tool in Maestro for fingerprinting computations [33]. We selected the receptor–ligand complexes, recorded any contact, aligned the sequences since they were all distinct, maintained all advanced settings as default, and generated the fingerprints. In the computed matrix, we checked for any contact and coloured the main plot in order to show the receptor–ligand interactions, only keeping the interacting residues. We also kept two plots of the ligand interactions and the residue interactions.

### 4.4. Pharmacokinetic and DFT Studies

Pharmacokinetic studies examine how the body absorbs, distributes, metabolises, and excretes drugs, and help to understand a drug’s processing in the body, bioavailability, metabolism rates, and elimination. These studies guide drug dosing, ensuring safe and effective therapeutic outcomes in medical treatment. The pharmacokinetic studies of the compounds were performed during the screening with the QikProp tool and Lipinski’s rule was kept as a filter, which is a significant part of this study [33,47,48]. Standard values were employed for descriptor comparison. Density Functional Theory (DFT) optimization was executed using the Jaguar program in Maestro, applying quantum mechanics for refining the molecular structures and exploring electronic properties [33,55]. In the context of drug design, Jaguar aids in improving the accuracy of molecular representations, helping to understand the energetics and behaviour of molecules and also enhancing the reliability of subsequent computational analyses and predictions [33,56]. In the input, ligand molecules were considered using the default B3LYP-D3 theory with a 6–31 G** basis set. DFT theory was chosen in the theory tab, maintaining automatic SCF spin treatment. The three-bond dispersion correction with all relevant dispersion-corrected functions was enabled. The quick accuracy level was retained in the SCF tab, initialised with an atomic overlap guess. The convergence criteria included a maximum of 48 iterations, an energy change of 5 × 10^−5^ Hartree, and an RMS density matrix change of 5 × 10^−6^. This configuration ensured precise and efficient quantum calculations for the given molecular system [33,55]. The SCF convergence methods included a level shift of 0 Hartree, maintaining no thermal smearing, and employing the DIIS convergence scheme. Consistent orbital sets were utilized for isomeric input structures. All input structures adhered to a unified basis set with no final localization. In the optimisation tab, a maximum of 100 steps were kept and switched to analytic integrals near convergence. The convergence criteria were kept default, and the initial hessian of the Schlegel guess was kept with coordinates of redundant internals. In the properties tab, we kept the vibrational frequencies using the available Hessian, IR intensities, and most abundant isotopes, while the thermochemistry was kept at 1.0 atm pressure with a starting temperature of 298.15 K. We computed surfaces (molecular orbitals, density, and potential) for electrostatic potential, average local ionization energy, non-covalent interactions, electron densities, spin density, HOMO, and LUMO; the PBF solvent model with water as the solvent was maintained in solvation. Output files were saved for analysis using the QM-Monitor tool [33,55].

### 4.5. Molecular Dynamics Simulation’s System Preparation and Production Run

In this study, we used the Desmond-based molecular dynamics simulation and the interaction dynamics of IRESSA with breast cancer-related proteins that provided intricate details on protein–ligand interactions, offering insights into stability, flexibility, and intermolecular forces [33,57]. This computational approach enhances our comprehension of IRESSA’s behaviour at the atomic level, informing drug design strategies [58]. The system builder and molecular dynamics are tools in computational chemistry that construct initial structures for molecular simulations and involve production runs by simulating the movement of atoms and molecules over time [33,57,58]. Together, they enable the study of the dynamic behaviours in complex molecular systems. We used the system builder tool to prepare the system file, where we used the predefined SPC water model with the boundary conditions of the orthorhombic box shape in a buffer medium of 10 × 10 × 10 Å distance, and investigated if it fits properly on the P–L complexes [59]. The ion and slat placement within 20 Å was excluded. The ions were neutralised by adding 2Cl^−^ in the IRESSA–4KD7 complex, 6Na^+^ in the IRESSA–3RCD complex, 5Na^+^ in the IRESSA–1M17 complex, and 10Na^+^ in the IRESSA–5NWH complex. The volume was also minimised so that it can fix appropriately onto the P–L complex. The OPLS4 forcefield [41,42] was also used, which has resulted in 23,913 atoms for the IRESSA–4KD7 complex, 35,222 atoms for the IRESSA–3RCD complex, 57,074 atoms for the IRESSA–1M17 complex, 29,060 and atoms for IRESSA–5NWH complex. We used the Molecular Dynamics Panel to load the prepared complexes into the system builder, and we set the simulation time to 100 ns with a trajectory recording interval of 100 ps and an energy level of 1.2, resulting in 1000 frames for each condition [33,57,58]. The NPT ensemble class was applied at a temperature of 300 K and a pressure of 1.01325 bar. The system was relaxed before production, and the simulation’s trajectory file was examined using the Simulation Interaction Diagram tool [33,60].

## 5. Conclusions

Breast cancer remains a significant global health concern, impacting millions of women globally. Our study highlights the complex challenge of drug resistance in breast cancer, emphasising the need for innovative solutions. Our multitargeted screening identified IRESSA as a promising inhibitor against crucial proteins. The interaction fingerprints, DFT, ADMET, and molecular dynamic simulations demonstrated its stability, suggesting potential efficacy. While IRESSA is FDA-approved for lung cancer, its multitargeted properties make it a candidate for repurposing for use against breast cancer. Experimental validation is crucial to confirm its role, but these findings offer a promising avenue for further research in combating breast cancer drug resistance.

## Figures and Tables

**Figure 1 pharmaceuticals-17-00208-f001:**
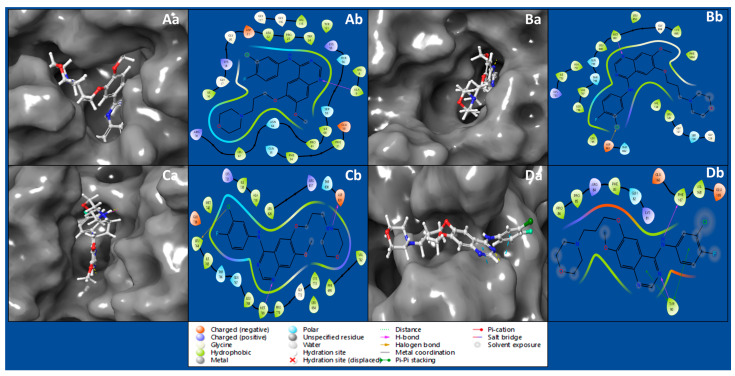
Showing the ligand interaction diagram of proteins with ligand IRESSA to show the coverage on the pocket and a detailed interaction view of (**Aa**) 4KD7 in 3D, (**Ab**) 4KD7 in 2D, (**Ba**) 3RCD in 3D (**Bb**) 3RCD in 3D (**Ca**) 1M17 in 3D and (**Cb**) 1M17 in 2D, and (**Da**) 5NWH in 3D (**Db**) 5NWH in 2D. Furthermore, the legend is provided to understand the residue and interaction types.

**Figure 2 pharmaceuticals-17-00208-f002:**
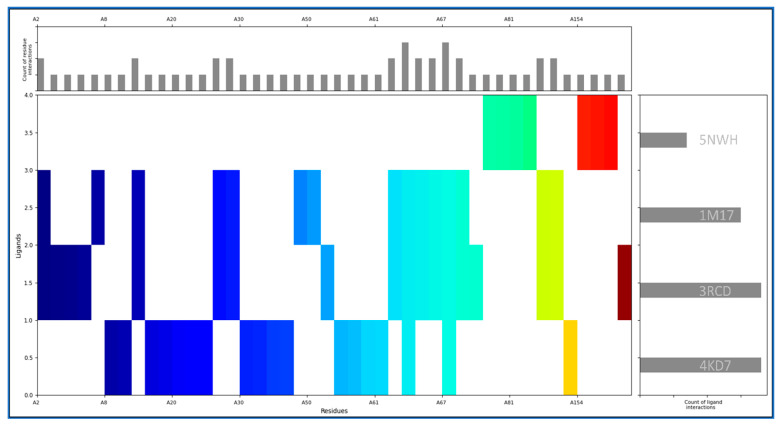
Showing the Molecular Interaction Fingerprinting of IRESSA with all four proteins. The coloured plot shows the interacting residue distributions, the count of ligand interactions in the right-side plot, and the count of residue interactions in the upper-side plot in order to understand which residue and ligand form the most interactions.

**Figure 3 pharmaceuticals-17-00208-f003:**
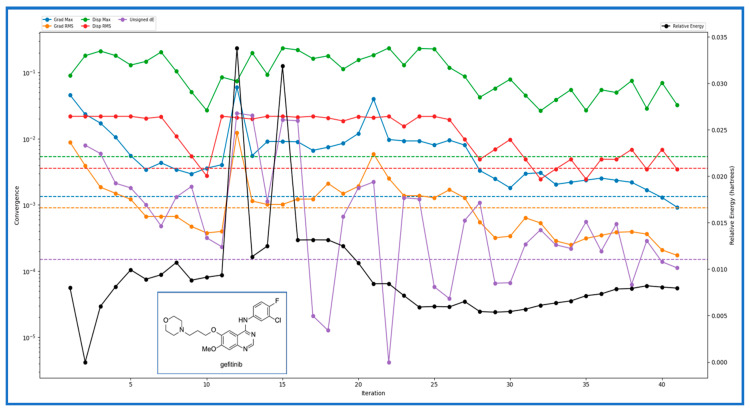
Showing the different energy levels generated after the various iterations over time and compared with the relative energy levels of the compounds. Blue shows the Grad Max, green shows the Disp Max, orange shows the Grad RMS, and red shows the Disp RMS. The Unsigned dE is magenta, while the relative energy (Hartree) is shown in Black.

**Figure 4 pharmaceuticals-17-00208-f004:**
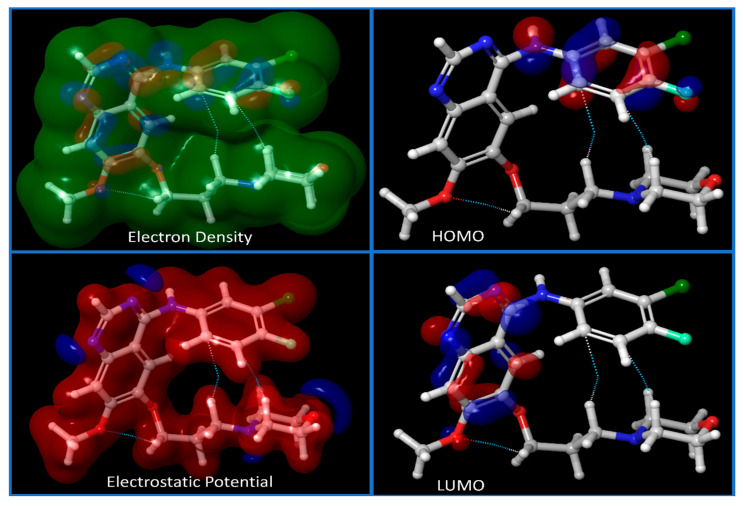
Showing the different energy levels of the compound IRESSA. We have shown the Electron Density, Electrostatic potential of the compound, and HOMO and LUMO sides of the compound to understand its energy level at lower and higher sides.

**Figure 5 pharmaceuticals-17-00208-f005:**
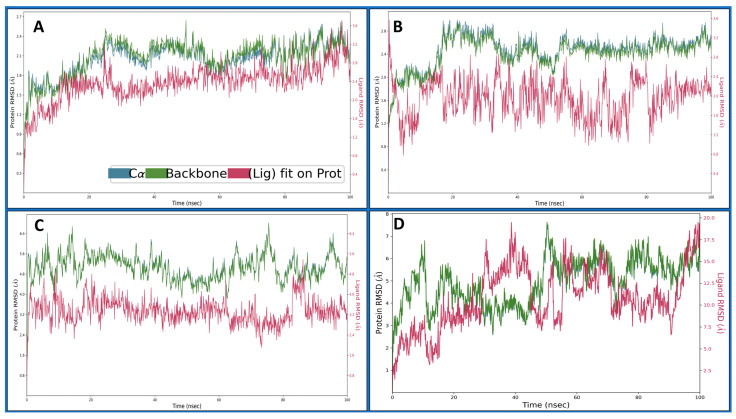
Showing the Root Mean Square Deviation (RMSD) of IRESSA (red) in complexes with Cα (Blue) and Backbone (army green) of the proteins of (**A**) 4KD7, (**B**) 3RCD, (**C**) 1M17, and (**D**) 5NWH during the 100 ns MD Simulation.

**Figure 6 pharmaceuticals-17-00208-f006:**
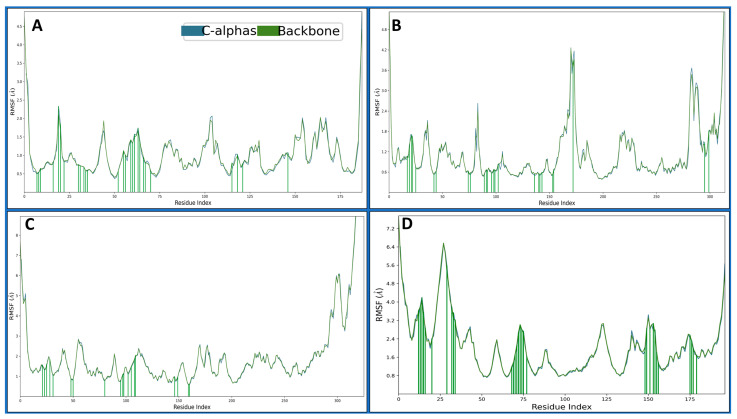
Showing the Root Mean Square Fluctuations (RMSF) Cα (Blue) and Backbone (army green) of the proteins of (**A**) 4KD7, (**B**) 3RCD, (**C**) 1M17, and (**D**) 5NWH. The green lines show the ligand interactions during the 100 ns MD simulation.

**Figure 7 pharmaceuticals-17-00208-f007:**
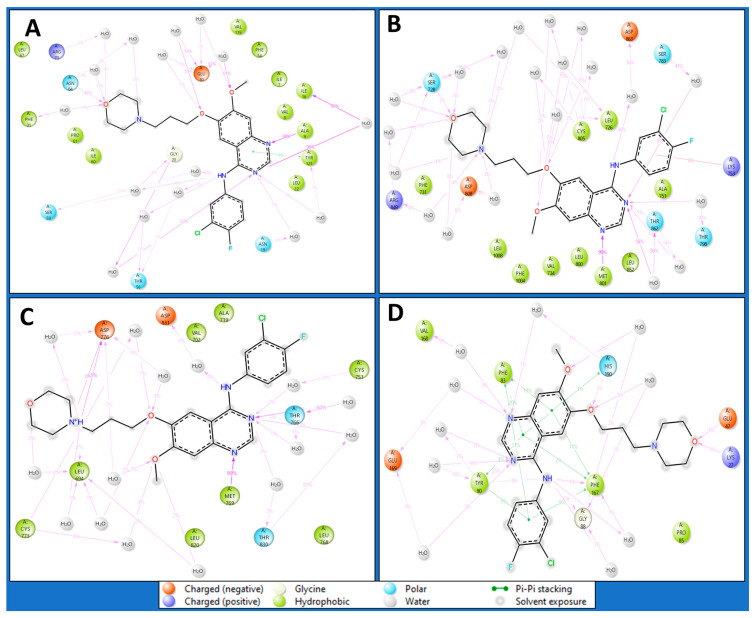
Simulation Interaction Diagram of IRESSA in complexes with (**A**) 4KD7, (**B**) 3RCD, (**C**) 1M17, and (**D**) 5NWH. The legend is provided to understand the interaction and bond types.

**Figure 8 pharmaceuticals-17-00208-f008:**
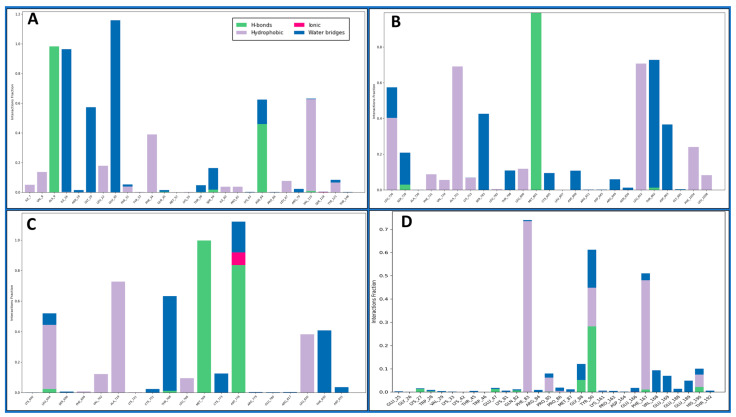
The count of the Simulation Interaction Diagram of IRESSA in complexes with (**A**) 4KD7, (**B**) 3RCD, (**C**) 1M17, and (**D**) 5NWH, where the H-bond is shown in green, ionic in red, hydrophobic in grey, and water bridges in blue.

**Figure 9 pharmaceuticals-17-00208-f009:**
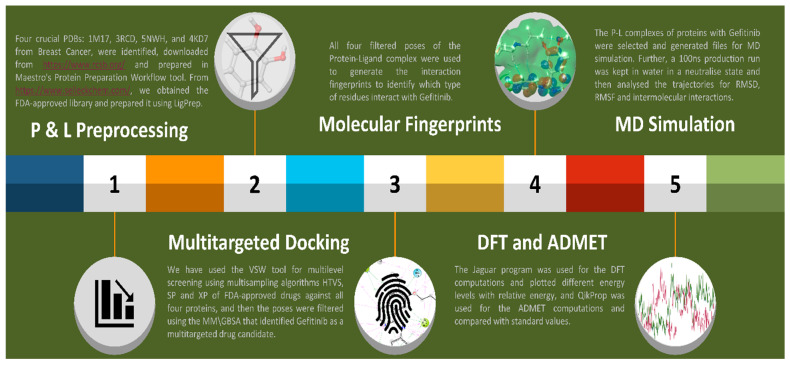
The Graphical Abstract shows a complete study and the methods followed, from data processing to MD simulation and reporting IRESSA against breast cancer.

**Table 1 pharmaceuticals-17-00208-t001:** Showing the docking (Kcal/mol) and MM/GBSA scores (Kcal/mol) along with other computations among all four proteins and the IRESSA drug.

S No	PDB ID	Docking Score	MMGBSA	Prime Hbond	Prime vdW	Ligand Efficiency ln	Ligand Efficiency sa
1	4KD7	−8.809	−59.08	−91.21	−897.44	−1.987	−0.893
2	3RCD	−8.459	−60.59	−152.23	−1316.69	−1.908	−0.857
3	1M17	−9.021	−61.74	−151.3	−1374.68	−2.035	−0.914
4	5NWH	−4.527	−49.09	−97.75	−703.04	−1.021	−0.459

**Table 2 pharmaceuticals-17-00208-t002:** Showing the ADMET properties of the IRESSA Drug and its comparison with the standard values of QikProp.

Descriptors	Iressa	Standard Values	Descriptors	Iressa	Standard Values
#acid	0	0–1	HumanOralAbsorption	3	-
#amide	0	0–1	IP(eV)	8.475	7.9–10.5
#amidine	0	0	Jm	0.007	-
#amine	1	0–1	mol MW	446.908	130.0–725.0
#in34	0	-	PercentHumanOralAbsorption	100	>80% is high, <25% is poor
#in56	22	-	PISA	242.502	0.0–450.0
#metab	5	1–8	PSA	61.141	7.0–200.0
#NandO	7	2–15	QPlogBB	0.312	−3.0–1.2
#noncon	4	-	QPlogHERG	−7.087	concern below −5
#nonHatm	31	-	QPlogKhsa	0.349	−1.5–1.5
#ringatoms	22	-	QPlogKp	−2.682	−8.0–−1.0
#rotor	8	0–15	QPlogPC16	13.202	4.0–18.0
#rtvFG	0	0–2	QPlogPo/w	4.31	−2.0–6.5
#stars	0	0 – 5	QPlogPoct	20.444	8.0–35.0
accptHB	7.7	2.0–20.0	QPlogPw	10.783	4.0–45.0
ACxDN^.5/SA	0.0101519	0.0–0.05	QPlogS	−5.129	−6.5–0.5
Category	small	-	QPPCaco	1049.999	<25 poor, >500 great
CIQPlogS	−5.22	−6.5–0.5	QPPMDCK	2306.642	<25 poor, >500 great
CNS	1	−2 (inactive), +2 (active)	QPpolrz	44.448	13.0–70.0
dip^2/V	0.0220798	0.0–0.13	RuleOfFive	0	maximum is 4
dipole	5.429	1.0–12.5	RuleOfThree	0	maximum is 3
donorHB	1	0.0–6.0	SAamideO	0	0.0–35.0
EA(eV)	1.279	−0.9–1.7	SAfluorine	41.345	0.0–100.0
FISA	39.187	7.0–330.0	SASA	758.477	300.0–1000.0
FOSA	366.922	0.0–750.0	volume	1334.913	500.0–2000.0
glob	0.7730209	0.75–0.95	WPSA	109.866	0.0–175.0

## Data Availability

The manuscript provides all the data in figures and tables.

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
