# Peer review of "Structure-Based In Silico Approaches Reveal IRESSA as a Multitargeted Breast Cancer Regulatory, Signalling, and Receptor Protein Inhibitor"

_pharmaceuticals, 2024, doi:10.3390/ph17020208_

Round 1

Reviewer 1 Report

Comments and Suggestions for Authors

REVIEW REPORT 1

Journal: Pharmaceuticals (ISSN 1424-8247)

Manuscript ID: pharmaceuticals-2841339

Title: Structure-based in-silico approaches reveal Iressa as a multitargeted breast cancer regulatory, signalling, and receptor protein inhibitor.
Authors: Hassan Hussain Almasoudi, Mutaib M Mashraqi , Saleh A. Alshamrani, Afaf Alharthi, Ohud Alsalmi, Mohammed H Nahari, Fares Saeed H Al-Mansour, Abdulfattah Y. Alhazmi *

The manuscript describes multitargeted screening of FDA- approved for Lung cancer drug IRESSA™ (Gefitinib) and pinpointed it as a promising inhibitor against crucial proteins of breast cancer malignances. My first comment is that this research is developed in a current scientific field - “drug repurposing”, which is with great potential for future work. The outlined idea, supported with a well-presented graphical abstract, for comprehensive screening has been performed precisely and qualitatively by means of a large set of analyses, which makes a very good impression. However, my further comments, some suggestions and questions to authors, which may help to improve this report, are listed below:

1.      I recommend an additional punctuation and grammar check of the manuscript.

2.      In my opinion the manufacturer name of Gefitinib is capitalized letters – not Iressa, but IRESSA™. It should be checked and if necessary edited in the hole manuscript.

3.      Into the “Discussion” section (Line 552 – 574) I found explanation why the authors conduct their research around drug IRESSA™. There are pointed some citations (number 58 -60) with relevant contemporary applications and investigations on Gefitinib. Nevertheless, I haven’t found information into the “Introduction” section, where the main accent of the proposed research must be well-explained. In the literature it’s easy to find a number of investigations of IRESSA™ efficiency in breast cancer (including systematic reviews, meta-analysis and even Phase II clinical trials). It should be explained in “Introduction” section, too.

4.      Do you have data for structure-based in silico-approaches for nanomedicines and IRESSA™ (“hot topic” in personalized medicine approaches in breast cancer)? Some future investigations in this direction also could be valuable scientific knowledge.

Reviewer Conclusion: After the editing of the proposed minor revisions about the above suggestions for improvement of the manuscript I conclude that this paper can be published in the Journal Pharmaceuticals as a valuable scientific report.

Author Response

Journal: Pharmaceuticals (ISSN 1424-8247)

Manuscript ID: pharmaceuticals-2841339

Title: Structure-based in-silico approaches reveal Iressa as a multitargeted breast cancer regulatory, signalling, and receptor protein inhibitor.
Authors: Hassan Hussain Almasoudi, Mutaib M Mashraqi , Saleh A. Alshamrani, Afaf Alharthi, Ohud Alsalmi, Mohammed H Nahari, Fares Saeed H Al-Mansour, Abdulfattah Y. Alhazmi *

The manuscript describes multitargeted screening of FDA- approved for Lung cancer drug IRESSA™ (Gefitinib) and pinpointed it as a promising inhibitor against crucial proteins of breast cancer malignances. My first comment is that this research is developed in a current scientific field - “drug repurposing”, which is with great potential for future work. The outlined idea, supported with a well-presented graphical abstract, for comprehensive screening has been performed precisely and qualitatively by means of a large set of analyses, which makes a very good impression. However, my further comments, some suggestions and questions to authors, which may help to improve this report, are listed below:

Thank You so much for your valuable suggestions and appreciation of our manuscript. 

1. I recommend an additional punctuation and grammar check of the manuscript.

Dear Reviewer, We have used the Grammarly for checkign the mistakes and we corrected it.

2. In my opinion the manufacturer name of Gefitinib is capitalized letters – not Iressa, but IRESSA™. It should be checked and if necessary edited in the hole manuscript.

Dear Reviewer, we now have checked and corrected as you suggested. 

3.  Into the “Discussion” section (Line 552 – 574) I found explanation why the authors conduct their research around drug IRESSA™. There are pointed some citations (number 58 -60) with relevant contemporary applications and investigations on Gefitinib. Nevertheless, I haven’t found information into the “Introduction” section, where the main accent of the proposed research must be well-explained. In the literature it’s easy to find a number of investigations of IRESSA™ efficiency in breast cancer (including systematic reviews, meta-analysis and even Phase II clinical trials). It should be explained in “Introduction” section, too.

Dear Reviewer, we have now updated the introduction as you suggested. 

4.      Do you have data for structure-based in silico-approaches for nanomedicines and IRESSA™ (“hot topic” in personalized medicine approaches in breast cancer)? Some future investigations in this direction also could be valuable scientific knowledge.

Dear Reviewer, we have not explored nanomedicine as of now. However, we will explore it in coming future. 

Reviewer Conclusion: After the editing of the proposed minor revisions about the above suggestions for improvement of the manuscript I conclude that this paper can be published in the Journal Pharmaceuticals as a valuable scientific report.

Dear Reviewer, all the authors are extremely thankful to you for providing valuable points and suggestions to improve the manuscript. 

Reviewer 2 Report

Comments and Suggestions for Authors

Although the abstract is well written, it might have to be shortened, the authors should check the maximum amount of words allowed by MDPI; I suggest shortening the introductory section, in order to leave the results;

- The molecular action mechanisms of the studied compound should be schematized and a figure regarding this should added, in order for the reader to have a better visual understanding of the matter;

- Figure quality (resolution) should be improved (most times words are a bit hard to read);

- The potential administration route and formulation type and composition for the studied compound should be mentioned;

- Whether the authors consider these results as good enough so that Iressa could be used in the future as primary cancer therapies, or just as adjuvant therapies, should be further discussed;

- An abbreviation list is missing and should be added.

Author Response

Although the abstract is well written, it might have to be shortened, the authors should check the maximum amount of words allowed by MDPI; I suggest shortening the introductory section, in order to leave the results.

Dear Reviewer, we have already provided the shortest abstract possible, and the MDPI team has formatted the manuscript accordingly. 

  • The molecular action mechanisms of the studied compound should be schematized and a figure regarding this should added, in order for the reader to have a better visual understanding of the matter;

Dear Reviewer, we agree with your suggestions. However, our manuscript is for small molecule-based drug design. Adding molecular action mechanisms will take this in another direction. In future, while validating this compound in the experimental lab, we will add and keep your suggestion in mind. 

  • Figure quality (resolution) should be improved (most times words are a bit hard to read);

Dear Reviewer, we have provided 1200 DPI images to the journal to publish. 

  • The potential administration route and formulation type and composition for the studied compound should be mentioned;

Dear Reviewer, we agree with your suggestion and now we have added it in the discussion.  

  • Whether the authors consider these results as good enough so that Iressa could be used in the future as primary cancer therapies, or just as adjuvant therapies, should be further discussed;

Dear Reviewer, we agree with your suggestion. Now we have added this in the manuscript. 

  • An abbreviation list is missing and should be added.

Dear Reviewer, we have now provided the list of abbreviations in a single place to help the readers. 

Dear Reviewer, all the authors are extremely thankful to you for your suggestions and valuable comments.